# *Rigor in AI:* Doing Rigorous AI Work Requires a Broader, Responsible AI-Informed Conception of Rigor

**Alexandra Olteanu**[1*]     **Su Lin Blodgett**[1]     **Agathe Balayn**[1]     **Angelina Wang**[2]
**Fernando Diaz**[3]     **Flavio du Pin Calmon**[4]     **Margaret Mitchell**[5]
**Michael Ekstrand**[6]     **Reuben Binns**[7]     **Solon Barocas**[1]
[1]Microsoft Research     [2]Cornell Tech     [3]Carnegie Mellon University     [4]Harvard University
[5]Hugging Face     [6]Drexel University     [7]University of Oxford
[*]Corresponding author: alexandra.olteanu@microsoft.com

## Abstract

In AI research and practice, *rigor* remains largely understood in terms of *methodological rigor*—such as whether mathematical, statistical, or computational methods are correctly applied. We argue that this narrow conception of rigor has contributed to the concerns raised by the responsible AI community, including overblown claims about the capabilities of AI systems. Our position is that a broader conception of what rigorous AI research and practice should entail is needed. We believe such a conception—in addition to a more expansive understanding of (1) *methodological rigor*—should include aspects related to (2) what background knowledge informs what to work on (*epistemic rigor*); (3) how disciplinary, community, or personal norms, standards, or beliefs influence the work (*normative rigor*); (4) how clearly articulated the theoretical constructs under use are (*conceptual rigor*); (5) what is reported and how (*reporting rigor*); and (6) how well-supported the inferences from existing evidence are (*interpretative rigor*). In doing so, we also provide useful language and a framework for much-needed dialogue about the AI community's work by researchers, policymakers, journalists, and other stakeholders.

## 1   Rigor in AI Research and Practice

Rigor remains a subject of intense debate in science [e.g., 2, 20, 43, 53, 68, 81, 104, 107, 129], a debate we are unlikely to resolve here. We thus keep our goal relatively modest: to help broaden the AI community's perspective of what rigorous AI[1] work should entail. We argue this is critically needed as *relying on impoverished conceptions of rigor can have an undesirable, yet formative impact on the quality of both AI research and practice*—heightening concerns ranging from unsubstantiated claims about AI systems [e.g., 35, 115, 127, 141, 147] to a plethora of unintended consequences [e.g., 80, 110, 126].

**Rigor in AI:** The debate surrounding rigor in science has by no means evaded the AI community [e.g., 65, 87, 117, 130]. In AI research and practice, rigor remains largely understood in terms of *methodological rigor*, which is typically conceptualized as whether mathematical, statistical, or computational methods are correctly applied; whether new methods, models, or systems are tested on large-scale or complex benchmarks and compared with a sufficient number of competing methods, models, or systems; whether the methods or analyses can scale or generalize; or whether the phenomena under analysis were—in contrast to more qualitative work—mathematically formalized

---

[1]Throughout this paper, we deliberately leave the term AI under-specified in order to capture the full constellation of meanings the term has been used for, and thus any work that practitioners or researchers would describe as AI—i.e., if they consider themselves as working on AI, our call is for them. We thus mainly use the term as a modifier to help scope the community this call is addressing, and the work and artifacts this community does or uses.

| Facet of rigor | What the facet is concerned with | What the facet asks for |
|---|---|---|
| **Epistemic rigor (§2.1)** | What *background knowledge* informs which problems are addressed and how? | Is the *background knowledge* clearly and explicitly communicated? Is the *background knowledge* appropriate, well-justified, and appropriately applied? |
| **Normative rigor (§2.2)** | Which disciplinary, community, organizational, or personal *norms, standards, values, or beliefs* influence the work and how? | Are these *norms, standards, values, or beliefs* clearly and explicitly communicated? Are these *norms, standards, values, or beliefs* appropriate & appropriately followed? |
| **Conceptual rigor (§2.3)** | Which *theoretical constructs* are under investigation? | Are the *theoretical constructs* clearly and explicitly articulated? Are the *theoretical constructs* appropriate and well-justified? |
| **Methodological rigor (§2.4)** | Which *methods* are being used? | Are these *methods* and their use clearly and explicitly described? Are these *methods* appropriate, well-justified and appropriately applied? |
| **Reporting rigor (§2.5)** | What are the *research findings*? | Are the *research findings* clearly communicated? Is the presentation of *research findings* appropriate and well-justified? |
| **Interpretative rigor (§2.6)** | What *inferences* are being drawn from the research findings? | Are these *inferences* clearly and explicitly communicated? Are these *inferences* appropriate and well-justified? |

Table 1: Overview of the six facets of rigor in AI research and practice. For each facet, we highlight what the facet is concerned with (*descriptive* overview)—i.e., what the *objects of concern* is for the facet—and what the facet asks for (*evaluative* overview).

and quantified [e.g., 22, 26, 49, 60, 65, 75, 130, 131, 142, 156]. These conceptualizations are often shaped by implicit assumptions that more complex methods and architectures and larger data samples are better [e.g., 9, 47, 83, 139, 142], by the common practice of relying on benchmark-driven evaluations [e.g., 86, 144, 156], and by a dominant mode of thinking oriented towards algorithmic formalism (centered on ideas of objectivity/neutrality, abstract and mathematical representations, and universalism/generalization) [e.g., 22, 44, 60, 89, 149]. Yet such conceptualizations of rigor may fail to demand, for instance, that benchmarks be fit-for-purpose or proven to measure what they claim to be measuring [e.g., 28, 113, 145], or that the knowledge or assumptions the work relies on be reliable or valid [e.g., 28, 92, 98, 115]. This puts into question the integrity and reliability of any conclusions drawn based on such benchmark-centered evaluations or that depend on questionable assumptions.

While such failures threaten the scientific integrity of AI research, foreseeing and addressing the consequences of putting insufficiently rigorous AI artifacts (e.g., models, systems, applications, outputs, data) into practice are often seen as the purview of responsible AI—which is nevertheless generally seen as separate from rigor, as it is understood as more concerned with ethics, stakeholders, societal impacts and harms, and real-world deployment scenarios. However, it is often only in real-world deployment scenarios or when considering stakeholders that the inadequacies and impact of current approaches to (lack of) rigor in AI research and practice become clear. We thus recast this distinction, and argue that by making visible and demanding attention to such failures, *responsible AI asks researchers and practitioners to uphold principles of scientific integrity in their work.* Although responsible AI is often seen as out of scope for an AI researcher or practitioner not engaged in that space, *any scientist should see rigor as well within scope.* In other words, even though unrigorous AI work implicates what are often seen as responsible AI consequences, we argue that a broader notion of rigor that accounts for what produces these consequences needs to be considered by *all* AI researchers and practitioners.[2]

We take the position that **doing rigorous AI work requires broadening our understanding of what rigor in AI research and practice should entail by drawing on work by the responsible AI community.**[3] To this end, we foreground a wider range of critical considerations from responsible AI literature (broadly construed) for rigorous AI work. Instead of prescribing rigid criteria for rigorous AI work, our goal is to provide useful scaffolding for much-needed dialogue about and scrutiny of the community's work by researchers as well as policymakers, journalists, and other stakeholders.

## 2   Facets of Rigor in AI Research and Practice

We foreground six key facets of rigor—*epistemic*, *normative*, *conceptual*, *methodological*, *reporting*, and *interpretative*—that we argue AI research and practice should contend with. While it may be

---

[2]*Positionality statement:* In recent years, we have noticed increasing tensions within and across AI and AI-adjacent communities concerning what research questions should be prioritized, who can even be trusted to conduct certain types of work, and what rigorous research entails. We also observed trends in the types of work submitted and published at both AI and AI-adjacent venues, raising concerns about a pernicious lack of conceptual clarity, growing pockets of research motivated by hypothetical, speculative settings, and an increasing use of ambiguous, non-productive terminology. These experiences inform and motivate this paper. Our perspective is also informed by our diverse disciplinary backgrounds spanning computer systems, theoretical and applied ML, information retrieval, computational social science, natural language processing, computational linguistics, computer vision, human-computer interaction, law and policy, science and technology studies, and philosophy.

[3]We use *responsible AI* to broadly refer to critical work on the impact of AI artifacts on people and society from several communities, including ethical AI, responsible AI, ethics of AI, and science and technology studies.

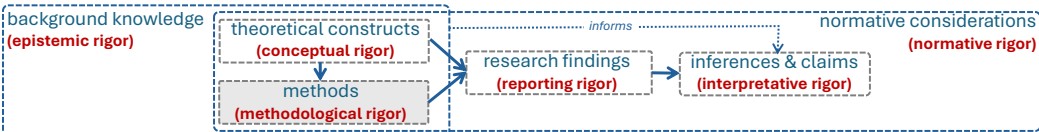

Figure 1: Simplified overview of the objects of concern for each **facet of rigor** and of common dependencies among them. For instance, the research findings typically determine what inferences and claims can be made, while normative considerations may influence the choices of theoretical constructs, of what methods to use, of what research findings to report, or of what inferences and claims to make. Dependencies are illustrated through both arrows as well as nested boxes.

difficult to draw clear boundaries between some of these facets, we discuss them separately to provide distinct lenses for reflecting on and interrogating the quality and scientific integrity of AI work (see Table 1). Specifically, for each facet, we provide a *descriptive overview*—what the facet is concerned with, i.e., what the *object of concern* is for that facet—and an *evaluative overview*—what the facet asks for. All facets of rigor are inherently about the *choices* we make about an *object of concern* (e.g., *epistemic rigor* is concerned with *background knowledge*, while *conceptual rigor* is concerned with *theoretical constructs*). Having these distinct facets encourages researchers and practitioners to think carefully about the *choices* they make for each *object of concern*, and disentangles possible debates about these choices—e.g., it separates debates about which *norms* we should follow (*normative rigor*) from what *background knowledge* should inform the work (*epistemic rigor*). For each facet, we discuss examples of mechanisms that help promote rigor along that facet, which can include a mix of processes and desiderata. While we foreground each mechanism for only one of the facets, we note that some mechanisms might help foster rigor along more than one facet (e.g., engaging with construct and internal validity concerns can foster both *methodological* and *interpretative rigor*).

Figure 1 provides a simplified overview of the *objects of concern* for each facet of rigor and of possible common dependencies among them. It illustrates how limiting our conception of rigor to methodological concerns may obfuscate how *our work and the claims we make are shaped by a variety of choices that both precede and succeed any methodological considerations*, even when some of those choices remain tacit or implicit (e.g., as is often the case for common disciplinary norms, standards, or practices [56, 156]). The figure also underscores how it may be difficult to make good choices for "downstream" objects when "upstream" objects are poorly chosen. For instance, making poor construct choices (*conceptual rigor*) may reduce our chances of operationalizing those constructs well (*methodological rigor*). The different facets of rigor may, however, also be tangled in complex relations of mutual interdependency [e.g., 44, 57, 145]—as methodological choices may, for instance, in turn "limit the structure of one's theoretical con[structs]" [44] if the methods determine what is observed and thus theorized about—and present choices may (and usually do) impact future work. For example, a lack of *interpretative rigor*—when ambiguous or baseless claims are being made—can have *epistemic* consequences for future work relying on those claims [e.g., 41, 65]. We further unpack these below:

## 2.1 Epistemic rigor

*What background knowledge informs which problems are addressed and how? Is this background knowledge clearly and explicitly communicated? Is the background knowledge appropriate, well-justified, and appropriately applied?*

Before considering any methodological questions, we have to contend with what it is that is being investigated, why it merits consideration, and what knowledge is "not under investigation, but [is] assumed, asserted, or essential" [61]. That is, we need to clarify the facts and background assumptions upon which the work relies in order to establish what the foundation for the work is and whether that foundation is sound. A frequently given example for epistemic failures is work asserting that records of physical appearance (e.g., photographs of faces) can be used to predict latent character traits (e.g., sexuality, political ideology, or criminality) [10]. Such work assumes that actions (what one does) or inner character (what one likes, thinks, or values) can be predicted based on physical attributes (how one looks). Although this assumption has its roots in "physiognomy"—regarded as pseudo-scientific as it relies on a set of epistemically baseless and extensively debunked claims [5, 137]—pockets of AI research recurrently draw upon it [5, 10]. This example also illustrates a broader class of epistemic failures: cases where a failure to scrutinize background assumptions may lead to work whose validity depends on whether existing methods, tools, or systems

function—or can be made to function—on given tasks when they do not or cannot [115], including because the tasks are conceptually or practically impossible or because there is no reliable evidence that those methods, tools, or systems are fit-for-purpose or reliable [98, 115, 119, 156].

Epistemic rigor is thus not only about making sure that the background knowledge new work builds on and how that knowledge is acquired are appropriate and appropriately applied, but also that that knowledge is appropriately justified [36]. Rigorously applying statistical or computational methods (*methodological rigor*) will do little to mitigate epistemic concerns if the problems being tackled or the assumptions underpinning the use of some methods are baseless, theoretically implausible, nonsensical, or grounded in pseudo-scientific or scientifically shallow work [3, 10, 80, 115, 133]. Why and what is being studied, built, or deployed [15]; why, what and whose problems are being prioritized [23, 124]; what implicit or tacit assumptions are being made about the stated problems or solutions under consideration [3, 93, 115]; and whether the choices of problems and methods are drawing on valid and well-founded evidence [24] are all examples of the types of key epistemic considerations that we should engage with. By contrast, easily available or common artifacts (such as datasets, methods, tools, or systems) tend to serve as "tools of opportunity, not instruments of epistemic rigor" [153], with researchers and practitioners often prioritizing work based on whether there are some existing artifacts available without appropriately scrutinizing the knowledge and assumptions underpinning them [156].

*Epistemic rigor, however, does not necessarily require specific epistemological commitments or choices but rather that those commitments and choices be made explicit.* While doing so may not definitively answer whether some problems or assumptions are baseless, nonsensical, or unethical, articulating epistemic commitments and the existing knowledge the work relies on lays down the grounds on which people can have discussions and work out disagreements [32].

**Mechanisms to promote *epistemic rigor*:** Ensuring work is *appropriately grounded* in past literature [e.g., 10] and that underlying assumptions are made explicit and *appropriately interrogated* [e.g., 93, 115, 156] can help foster epistemic rigor, and in turn all other facets of rigor (§2.2–§2.6).

*Appropriate grounding:* Common goals in AI research often include producing new insights, theories, or artifacts, providing evidence in support of existing theories, evaluating existing artifacts, or debunking prior work. A failure to review, appropriately situate within, and acknowledge prior literature, and how it influenced the questions being asked or the solutions being considered, can cast doubt on whether any meaningful progress towards those goals was actually made [e.g., 63, 68]. When such failures become systematic, research communities risk "curl[ing] up upon themselves and becom[ing] self-referential systems that orient more [internally]" [37] and developing their separate "terminology, source texts, and knowledge claims" [7]. Indeed, concerns that "[f]indings within [a research] community are often self-referential and lack [quality]" [67] are critical as this can make it difficult to trace the *provenance of central claims* that form the foundation that new work builds on or is motivated by. It can also result in an overall narrowing of what questions AI research tackles and the epistemic marginalization of certain viewpoints [6, 82, 98]. Appropriate grounding requires due diligence when reviewing and acknowledging work in one's subfield and related fields [18, 68, 101].

*Interrogating assumptions:* Any research work necessarily relies on assumptions about what is important, what is possible, or what represents sufficient or useful evidence [e.g., 118]. Not only does a lack of epistemic rigor make it hard to situate new work and the knowledge it produces in the context of existing knowledge it may draw from, relate to, corroborate, or dispute, but a lack of due diligence about the background knowledge underlying the work further risks bringing to the fore long-debunked claims from other disciplines—e.g., risking the reanimation of physiognomic methods [5, 10, 137]. Interrogating the assumptions underpinning such tasks—e.g., why would outer attributes be useful proxies for someone's character?—can expose them as a category error where observable traits (e.g., skin tone) are incorrectly treated as direct indicators of internal states (e.g., personality). Making background assumptions explicit thus facilitates our ability to scrutinize them [14, 156].

## 2.2 Normative rigor

*Which disciplinary, community, organizational, or personal norms, standards, values, or beliefs influence the work and how? Are these norms, standards, values, or beliefs clearly and explicitly communicated? Are these norms, standards, values, or beliefs appropriate and appropriately followed?*

Because of differences in underlying evidentiary standards, background assumptions, goals, ideals, and theoretical frameworks, different disciplines and communities have different ways of determining

what methods to use, what types of evidence are needed or sufficient, and why the resulting work or resulting knowledge matters [31, 36]. The assumptions and theories that underpin our work are not derived "out of thin air, but as a function of our experiences with the world in general and specifically with our colleagues, through dialogue, and the literature" [61], all of which can shape the type, direction, and quality of research [44, 57, 69]. Disciplinary and community norms are intertwined with a researcher's personal norms and drivers [45], e.g., what catches their interest or advances their goals. Making explicit how this mix of influences shapes the AI community's work can help others understand them and enable debates on their appropriateness.

Consider the emerging research on developing AI personas as tools to simulate users and human study participants [e.g., 8, 54, 84, 111]. While this research reflects common norms and beliefs in AI communities around scalability and efficiency [e.g., 4, 22], by aiming to replace human participants, such tools may fail to align or may even conflict with foundational values and norms around representation, participation, inclusion, and understanding [4, 148] that underpin many types of work this research seeks to support, such as user experience or social science studies with human participants. Such value conflicts "cannot be alleviated with better training or improved model performance alone" [4].

Common beliefs and expectations in the AI community—such as the goal of producing generalizable findings that are often abstracted away from any use context [22, 70]—have led to a reduction of problems and use scenarios "to a common set of representations or affordances" [22], and thus (even if implicitly) to de-valuing how context shapes datasets and tools. Reliance on undisclosed normative assumptions about problem statements, artifacts, or constructs that are contested or value-laden, yet treated as if they are neutral or generally applicable, can however be particularly misleading when a "field employs the language of procedural adherence to project a sense of certainty, objectivity, and stability" [59]. While dominant values in AI work like performance, generalization, and efficiency [22] may encourage extensive evaluations across as many metrics, benchmarks, and baselines as possible [22, 156], these evaluations tell us little about when, whether, or which improvements are necessary, desirable, or translate to meaningful benefits when deployed. Such expectations and norms—along resource-related considerations and constraints such as about costs and strict timeliness [64, 154, 156]—nonetheless influence what type of work gets prioritized or rewarded [50].

**Mechanisms to promote *normative rigor*:** All research is shaped by normative considerations. Normative rigor asks us to make these considerations explicit—such as via *ethical* and *positionality* statements [110] or other disclosure practices—and can include aspects related to expectations around the *significance and impact* of research [e.g., 61, 107] or what *ethical norms* to follow [e.g., 109].

*Research significance:* Central to normative considerations are expectations about how our work and its outcomes should intervene in the world, or why the work matters and is appropriate. The goal of research is often to "generate knowledge that will have positive practical impact" [12] or that advances the field [107], with norms around what work is worthwhile driven by desires to reward such work [34, 150]. There is however also an increasing recognition that the appraisal of the work's quality and significance should also include the work's potential for harm and not only for benefits [12, 29, 61]. Such appraisal typically rests on claims about the possible impacts from either the work's process or its outcomes. This echoes the concept of *consequential validity* from social science measurement scholarship [99, 145], which asks us when determining a measurement instrument's (in)validity to consider the consequential basis both of its use and of possible inferences (along with actions those inferences may entail) [72, 99]. Reckoning with the impact of our work thus promotes not only discussions about whether current norms around what constitutes good work are appropriate, but also promotes methodological (§2.4) and interpretative rigor (§2.6).

*Positionality and ethical statements:* Researchers' personal, disciplinary, and institutional backgrounds, their lived experiences, and their goals motivate and shape how they approach their work. *Positionality statements* are a mechanism meant to make such considerations explicit in order to help others contextualize the research and research outcomes [85, 110]. Positionality, however, does not only encompass aspects related to researchers' beliefs and values, but also those related to what knowledge they draw on, how they know what they know, and how they make methodological choices [19]. Thus, it can also aid or compromise epistemic (§2.1) and methodological rigor (§2.4). *Ethical statements* can further complement positionality statements by foregrounding the ethical concerns researchers grappled with or mitigated before or while conducting the work [13, 110]. Yet this practice of disclosing whether such concerns were considered and how they shaped any methodological choices (if at all) remains inconsistently adopted across AI communities.

## 2.3 Conceptual rigor

*Which theoretical constructs are under investigation? Are these theoretical constructs clearly and explicitly articulated? Are these theoretical constructs appropriate and well-justified?*

Assume a researcher wishes to evaluate whether a model "hallucinates." In AI research, the construct of "hallucination" has, however, been used to refer to several distinct types of system behaviors [95], including cases of generating content which is nonsensical, which contains factual errors, which is not in the input data, which is not in the training data, or a mix of these behaviors. Further, all these different understandings of what it means for a model to "hallucinate" are markedly different from the more common use of the term that requires an ability to perceive, feel, or have sensory experiences [e.g., 40]; and the term can thus carry meanings incompatible with AI systems. To understand what a researcher's evaluation of whether the model "hallucinates" means, we thus need to know which conceptualization they use, and if that conceptualization is sensible.

While contested constructs—those that have competing or even conflicting definitions, like "hallucination," "value alignment," "AGI," or "human-likeness"—are increasingly common in AI research and practice, their definitions often remain elusive, ambiguous, or poorly specified [e.g. 25, 26, 28, 145]. *Without clarity about what specifically we are analyzing, measuring, or striving for, it can be hard to assess progress or make any useful or reliable claims.* Work can also rely on constructs that are inconsistent with any theoretical tradition, such as treating identity categories as fixed and objective rather than continuous and constructed [92] or more generally failing to recognize that some constructs are innately fluid, non-deterministic, and fuzzy [21]. Such a lack of clarity can hinder replicability and reproducibility or may facilitate speculative post-factum interpretations, yielding possibly unsound and unfounded claims, or a conflation of proxy measurements (which may or may not measure any version of the underlying construct) with the construct under analysis. Thus, a lack of conceptual clarity can also undermining epistemic (§2.1), methodological (§2.4), and interpretative rigor (§2.6).

**Mechanisms to promote *conceptual rigor*:** Conceptual rigor requires attention to *conceptual clarity*—which construct we are after and how it is defined; appropriate *conceptual systematization*—the process by which the definition is made specific; and *terminological rigor*—that the terms used to refer to a construct do not harbor meanings that can lead to a misinterpretation of what the construct is.

*Conceptual clarity:* A growing number of objects in AI research are ambiguous or poorly specified, or else are objects for which we lack consensus about what they are or what they are for. The examination by Saphra and Wiegreffe [125] of what is meant by "mechanistic interpretability" is instructive: not only does the term have multiple competing meanings, but those meanings also reflect distinct disciplinary orientations and epistemic origins; a lack of clarity about which meaning is under use can obfuscate not only what the work does, but also why and how the work is done. Similar critiques have been made about the lack of conceptual clarity about what unlearning [42], bias [27], model collapse [128], interpretability [88], or generalization [89] mean. While we see growing concerns about the lack of conceptual clarity surrounding many aspects of AI research, from how desired capabilities are described to what metrics to optimize for [28, 65, 73, 125, 145], these remain largely overlooked in discussions about research integrity and quality in AI.

*Conceptual systematization:* In practice, many constructs involve a "broad constellation of meanings and understandings" [1], and working with them requires making choices about which meanings to use, "narrowing [them] into an explicit definition" [145]. The process of conceptual systematization asks researchers to engage not only with a high-level construct in the abstract, but to grapple more concretely with what it means in the context of the work they are conducting and how it relates to empirical observations or other constructs. Conceptual systematization is a prerequisite for rigorous measurement, (computational) specification, empirical analysis, and theory development [e.g., 1, 65, 113, 145], and is thus a prerequisite for methodological rigor (§2.4).

*Terminological rigor:* Conceptual rigor depends on terminological choices and what those choices communicate. Many terms in AI often carry over meanings from the human realm or other disciplinary contexts that are incompatible with AI systems, and can mislead [26, 38, 89, 118], suggest "unproven connotations," or lead to "collisions with other definitions, or conflation with other related but distinct concepts" [89]. Blili-Hamelin et al. [26] note that "when researchers equate human faculties with model proxies [...] [t]his rhetorical move is enabled by using colloquial terms like 'imagination' without considering whether it corresponds to the human faculty," leading to inflated claims. Clear, precise language helps "dispel speculative, scientifically unsupported portrayals of [AI] systems, and support more factual descriptions of them" [39], and thus clear scientific communication.

## 2.4 Methodological rigor

*What methods are being used? Are these methods and their use clearly and explicitly described? Are these methods appropriate, well-justified, and appropriately applied?*

Rigor is often "conceptualized as the appropriate execution of [methods]" [107], with discussions about rigor in AI centering around methodological considerations, from data collection and analysis, to model training and tuning, to experimental practices [e.g., 65, 130, 132], and notions of *theoretical rigor* (of algorithmic and mathematical analysis) and *empirical rigor* (of statistical and experimental approaches). Theoretical rigor typically seeks precise problem formulation using well-defined mathematical notation, accompanied by results (e.g., propositions, theorems, lemmas) with correct proofs—e.g., a clear and correct sequence of mathematically derived steps that support the stated result. Empirical rigor, in turn, seeks comparison of a proposed algorithm with a sufficient number of alternative—often competing—approaches, ablation studies, and some form of statistical analysis (e.g., power analysis, significance tests, or simply error bars). Renewed calls for methodological rigor have often been motivated by reproducibility concerns [52, 76, 94], with checklists and documentation practices proposed as a way to support reproducibility and replicability by standardizing methodological choices, recording them, and making them explicit.

**Mechanisms to promote *methodological rigor*:** Methodological choices are shaped by considerations about how to operationalize what we know—e.g., the background knowledge—to substantiate existing knowledge or produce new knowledge or artifacts. As methodological concerns have been central to discussions about rigor in AI, here we only focus on foregrounding mechanisms for aspects of methodological rigor we believe deserve added attention, including *construct validity* [28, 73, 145] and the need for *methodological standards*, particularly for high-risk domains [151]. For more comprehensive discussions of methodological rigor we direct the reader to [e.g., 65, 76, 89, 130, 132, 145].

*Construct validity:* Many problems in AI research, such as assessments of systems and phenomena, are concerned with measurement [73, 103, 145]. Even when there is conceptual clarity, ensuring construct validity—i.e., that measurement instruments (e.g., benchmark metrics) appropriately capture the construct of interest (e.g., reasoning, understanding, values)—is foundational to meaningful measurement and thus methodological rigor. A growing body of work has proposed frameworks and best practices for assessing the validity of measurements [e.g., 73, 91, 109, 143] and illustrated that existing measurement instruments exhibit a range of concerns that threaten their ability to measure what they purport to measure [28, 62, 105]. For example, Northcutt et al. [105] show widespread label errors in benchmark test datasets which can "destabilize ML benchmarks," thereby "lead[ing] practitioners to incorrect conclusions about which models actually perform best in the real world."

*Compliance with methodological standards:* Establishing methodological standards often involves extensive, community-wide debates about what methods are appropriate and when. Petzschner [112] notes how the failure of ML models intended for medical settings "to generalize to data from new, unseen clinical trials [...] highlight[s] the necessity for more stringent methodological standards," particularly for high-risk settings [151]. This has precipitated calls for developing standards in health datasets in AI applications [11]. The standardization of information retrieval evaluation practices via NIST's Text Retrieval Conference (TREC) was fundamental in revitalizing the research community and impacted the development of web search engines [121]. However, even though established standards can help a research community promote more rigorous debates about methodological choices, they may not by themselves ensure that those choices are explicitly reported or reflected on; Geiger et al. [56] hypothesize "that in fields with widely-established and shared methodological standards, researchers could have far higher rates of adherence to methodological best practices [...] but have lower rates of reporting that they actually followed those practices." By the same token, not making a choice of methods and presenting a kitchen sink (of metrics, methods) [156] undermines critical engagement with why the methods are appropriate. Compliance with standards should include explicit reflections on methodological choices and their application, including aspects related to constraints that researchers and practitioners had to navigate, such as access to participants, computing, or other resources [110].

## 2.5 Reporting rigor

*What research findings are being reported? Are these research findings clearly communicated? Is the presentation of research findings appropriate and well-justified?*

The understanding of research findings depends on what is communicated about these findings and how. Reporting rigor is concerned with making sure research findings are clearly and appropriately

communicated and justified. For instance, assume a researcher wishes to compare the performance of different recommender systems. Even when reporting only aggregated results, multiple options are possible, including averaging across ratings (treating each rating as equally important regardless of which user provided it or what item it was provided for), users (treating each user equally by first computing performance at user level), or items (treating each item equally by first computing performance at item level).[4] Depending on the data distribution (e.g., number of ratings per item/user), these differences may lead someone to draw different or even contradictory conclusions about which model performs best [e.g., 108]. Some ratings may also be more difficult to predict than others [e.g., 120], and any aggregation can obfuscate where exactly the model fails or succeeds [33, 89, 120]. Further, even such simple aggregations can introduce tacit assumptions about what should be optimized for e.g., to ensure a good predictive performance across all users versus all items [33]. Making these choices explicit can facilitate others' understanding of what specifically is being reported and why.

Such choices of what findings to report, and how—much as with choices of research questions, theoretical constructs, or methods—are shaped by our beliefs, values, and preferences, as well as disciplinary norms and incentives. Negative results are, for instance, less likely to be reported or published [136, 152], potentially "lead[ing] others to develop overly optimistic ideas about scientific progress on a particular topic" [12], and what is reported may be cherry-picked, such as "uncommonly compelling examples to illustrate the output of generative models" [12]. And as the example above also illustrates, even for the same findings, how the findings are reported or what about the findings is reported matters—such as choosing to report inferential uncertainty ("how precisely we have estimated the average for each group," when interested in estimating aggregate outcomes) versus outcome variability ("how much individual outcomes vary around averages for each group") [155]. Communicating the former risks leading people to "overestimate the size and importance of scientific findings" [155].

Poor choices of how findings are reported can thus also undermine interpretative rigor (§2.6)—what inferences or claims are made—as any statement of findings inherently embeds some interpretations while possibly hindering others. While this makes it difficult to fully separate concerns about reporting rigor from those about interpretative rigor, we foreground them separately to help draw attention to the different choices that often can be made about which findings to report and how.

**Mechanisms to promote *reporting rigor*:** Pre-study practices around disclosing how a study would be run and what about it would be reported like *pre-registration* [e.g., 66, 106, 140], and around reporting more granular, *disaggregated results* [e.g., 16, 33, 102] can promote reporting rigor.

*Pre-registration and reporting practices:* Reflecting on what to report about a study before conducting it can mitigate concerns about making such choices post-factum by hypothesizing after the results are known or reporting only from positive results [78, 106, 122]. Pre-registration is considered the "practice of specifying what you are going to do, and what you expect to find in your study, before carrying out the study" [140]. Pre-registration is seen as a mechanism for promoting more reliable research findings by differentiating between *confirmatory*—where hypotheses are tested and pre-registration is required—and *exploratory research*—where hypotheses are generated and pre-registration may not be required [106, 135]. While pre-registration often relies on narrow notions of what constitutes exploratory research and might inappropriately situate confirmatory research as more reliable [51], as a mechanism it can be used not only to encourage reflection on what measures of success will be used and reported on but also on how the study is designed [66], including in exploratory settings.

*Disaggregated evaluations:* As illustrated by the earlier example, aggregate measurements and metrics can obscure rare phenomena and information about where systems or models tend to fail or succeed, and can mask important effects [16, 33, 120, 130]. To mitigate such concerns, many have called for reporting disaggregated evaluations [33, 70, 89, 102], which "have proven to be remarkably effective at uncovering the ways in which AI systems perform differently for different groups of people" [16] and deemed a "critical piece of full empirical analysis" [130]. While conceptually simple, "their results, conclusions, and impacts depend on a variety of choices" [16], and they require careful consideration and justification—including by engaging with domain experts [138]—of how different choices of why, when, what, and how to conduct and report on such evaluations shape what inferences can be drawn and their impact. This, however, can in turn help make such choices explicit and encourage debates about their appropriateness.

---

[4]While this may often be left implicit due to a failure to explicitly systematize what *performance* entails (§2.3), each option likely reflects a somewhat different underlying conceptualization of performance—e.g., average user, item, rating-level performance—with "[d]ifferent 'findings' created by different conceptualizations" [57].

## 2.6 Interpretative rigor

*What inferences are being drawn from the research findings? Are these inferences clearly and explicitly communicated? Are these inferences appropriate and well-justified?*

Assume an AI system achieves high accuracy on a benchmark designed to assess mathematical reasoning [e.g., 58, 123]. As Salaudeen et al. [123] note, based on the system's performance on the benchmark, both of these two different alternative claims could be considered: the system can "solve linear algebra questions from a textbook accurately" or the system "has reached human-expert-level mathematical reasoning." Reliably moving from performance on a benchmark to either of the two claims requires clarity about any background assumptions concerning the feasibility of an AI system reaching human-level reasoning abilities [e.g., 141] (epistemic and normative rigor, §2.1–2.2), about how both "mathematical reasoning" and "human-expert-level" are conceptualized (conceptual rigor, §2.3), about whether the benchmark actually measures mathematical reasoning ability (methodological rigor, §2.4), and about how findings were reported—e.g., do we know what the performance on linear algebra questions is? (reporting rigor, §2.5). Echoing Markham [97]'s observation, a pernicious trap is to believe that methods and evaluation practices "bestow a natural interpretive clarity and self-reflexive awareness on the researcher," and thus "scientists must acknowledge the social and interpretative character of scientific discovery" [24].

*Both making and understanding knowledge claims require interpretation.* There are often multiple perspectives through which empirical, experimental, or theoretical evidence could be interpreted, and these different perspectives may not only lead to different aspects being emphasized but may also lead to different or even contradictory conclusions being drawn. While *reporting rigor* (§2.5) is concerned with choices about what findings to report and how, *interpretative rigor* is concerned with the conclusions we draw from these findings, and thus with the choices we make when we move from findings to either some *descriptive* claims—e.g., the system solves a given task—or to some *prescriptive* or *normative* claims—e.g., the system should be used to replace humans. Such conclusions or claims rarely directly follow from findings, but require some interpretation and are shaped by choices and considerations related to all other facets of rigor (§2.1–2.4)—they are influenced and made in the context of background knowledge, the relationship with the theoretical constructs under use, and the methods used to produce the findings and their limitations. While critical to how any work ultimately intervenes in the world, how claims are arrived at is often overlooked in discussions about rigor in AI, with the interpretation of results—i.e., what they mean, what people should do next—being treated as *self-evident*. The criteria for interpretative rigor is also "not whether the same interpretation would be independently arrived upon by different" people, but rather that "based on the evidence provided, is a given interpretation credible" or if "given all the same source information, would the interpretation stand up to scrutiny as being a justified, empirically grounded, exposition of the phenomenon" [107].

**Mechanisms to promote *interpretative rigor*:** Documenting and justifying evidence informing or situating possible claims can promote interpretative rigor, like via *documenting AI artifacts* [e.g., 17, 55, 102] or by engaging with possible threats to *internal/external validity* [e.g., 87, 109, 134].

*Documenting AI artifacts, their limitations, and their impacts:* Transparency about any AI artifacts (e.g., datasets, models, systems) under use can facilitate others' understanding of what claims may or may not be possible by providing added context about their characteristics and intended uses. To help scaffold and promote more transparent reporting on AI artifacts, researchers have developed tools, resources, or what Boyd [30] terms "context documents" (e.g., for datasets [17, 55, 114], models [102], services [116]). Complementing these efforts, others have argued for and put forward practices for recognizing and disclosing the *limitations*—i.e., "drawbacks in the design or execution of research that may impact the resulting findings and claims" [134]—and *impacts*—i.e., actual or possible consequences from the research, development, deployment, or use—of AI artifacts [13, 29, 48, 74, 90, 96, 100, 110, 134]. Limitations, in particular, directly impact how research findings can be interpreted—a failure to recognize them can lead to unsubstantiated claims, while a failure to disclose them can further lead to misinterpretations or misuse of claims. Impacts, in turn, can further affect prescriptive and normative claims and their implications, such about how one should act given the research findings.

*Internal and external validity:* Interpretative rigor requires not only careful deliberation on what the claims are about, but also whether there is appropriate evidence to support the claims. Establishing whether the research findings constitute appropriate evidence requires engaging with both *internal validity*—whether there are unaddressed issues with study design or execution that may compromise the findings such as data leakage [e.g., 77] or improper baseline comparisons [e.g., 87]—and *external*

*validity*—whether the findings generalize to different settings such as from one dataset [e.g., 113] or construct [e.g., 146] to another. For broader discussions, see Olteanu et al. [109] and Liao et al. [87].

## Concluding Reflections

By making the case for better documentation [55, 102, 116], better evaluation practices [70, 145, 156], better development and deployment practices [14, 74, 115], and a better understanding of impacts and limitations [13, 29, 134], *responsible AI research asks for greater scientific rigor.* The AI community has too often cast responsible AI considerations as out of scope, but holds up *research rigor as a virtue*. In AI research and practice, however, rigor still remains largely understood in terms of *methodological rigor*. Nevertheless, this can have unintended consequences as, for instance, "if the methodology is considered to be the *sine qua non* of scientificity, as it usually is, then there will be enormous pressures for the structure of *all* theories to accommodate to the theoretical structure embedded in the methodology [... with each] embedded theory involv[ing] its own value hierarchy" (emphasis original) [44]. In our reconception of rigor in AI—which expands beyond merely *methodological rigor*—we reframe many calls of the responsible AI community as within scope of all AI researchers and practitioners.

We argue that rigor in AI research and practice means more than just *methodological rigor*, and in so doing we bring doing AI work *responsibly*—as it pertains to *epistemic, normative, conceptual, methodological, reporting,* and *interpretative* rigor—under the umbrella of an AI researcher and practitioner's responsibility. By calling attention to these different facets of rigor, we also hope to provide the AI community with useful language that can help researchers and practitioners raise, clarify, and examine a wider range of concerns about existing practices in AI work. Nevertheless, while a broader conception of rigor can improve research integrity and quality, rigor is not a panacea for all problems in AI research and practice. While the facets of rigor we foreground are foundational to good science and apply broadly, we grounded our discussion about these facets in critical work from the responsible AI community because that work discusses these aspects in the context of AI research and practice—which is what our call is concerned with. We also do not claim that these facets are all-inclusive, but rather that they help demonstrate how expanding our conception of rigor beyond methodological considerations can help contribute evidence that AI work is rigorous.

### Alternative Views on Rigor in AI Research and Practice

*AI work is already rigorous or rigorous enough:*  Some may view addressing methodological rigor concerns as sufficient for ensuring rigorous AI work; under this view, rigor equates to and is achieved by focusing on methodological concerns. As underscored throughout this paper, this view leaves unaddressed a range of concerns which have produced undesirable outcomes, including a reliance on pseudo-scientific assumptions [e.g., 137], treatment of social phenomena inconsistent with broader scholarly understanding [e.g., gender 46, 79], systems that are not fit-for-purpose or cause harm [e.g., 71, 115], and claims or use of language that impedes public understanding of AI [e.g., 39].

*Non-methodological concerns are outside the purview of AI work:*  This view may arise because AI researchers and practitioners may see such concerns as outside the scope of core AI work [156]; they may also not see themselves as well-suited to addressing such concerns, either because it would be difficult to acquire the expertise needed, or because some concerns ought instead to be addressed by subject matter experts (with whom they may not have the time, desire, or resources to engage). We agree that engaging with subject matter experts can be valuable, and believe that AI work has been strengthened when it has done so [145]. Nevertheless, since all work involves *choices* about what problems are important, why they are important, and what precise objects are under investigation, and since AI work often claims real-world impact in or relevance to particular domains, we argue that it is impossible to do any AI work that does not implicate epistemic, normative, or conceptual questions—and thus researchers and practitioners must grapple with these concerns explicitly rather than implicitly.

*All rigor concerns are validity concerns:*  Under this view, this presentation of rigor concerns simply reframes work already addressing well-understood validity threats in AI research [e.g., 73, 87, 109, 123, 145]. While some of the concerns and mechanisms we describe (e.g., construct clarity/validity, internal/external validity) appear in the literature on validity, many considerations, particularly those related to epistemic, normative, conceptual, and interpretative rigor, precede or are a prerequisite to questioning and establishing validity, and facilitate reflection about issues beyond validity. For example, a failure to interrogate one's epistemological and normative commitments may result in systems that operationalize a construct as defined, but whose definition has been contested.

## Acknowledgments

We thank Michael Veale and Hanna Wallach for early conversations that have motivated this paper. We are also grateful to the members of the STAC team at Microsoft Research NYC for their feedback.

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
