# OpenReview forum: "Rigor in AI: Doing Rigorous AI Work Requires a Broader, Responsible AI-Informed Conception of Rigor"
_NeurIPS.cc/2025/Position_Paper_Track — NeurIPS 2025 Position Paper Track_

### Official Review · Reviewer_2RoQ · 2025-07-10

**Significance:** 3
**Presentation:** 2
**Rating:** 6
**Confidence:** 4

**Summary:**

This paper talks about the concept of rigor for AI, considering six facets of epistemic, normative, conceptual, methodological, reporting, and interpretative. The paper draws from concepts well known in science, and argues that the concept of rigor is often defined very narrowly in the AI community. The paper aims to create new discussions within the community about the concept of rigorous work, and it explains how such broader view can help the community understand and evaluate the abilities of AI models and to set better-informed research priorities.

**Strengths:**

Paper is well written.

Topic is timely and relevant.

If tone of the paper is changed, I think it can create very good discussions.

**Weaknesses:**

While I agree with most of what is proposed in the paper about promoting and upholding rigorous standards in various aspects of AI research and practice, I fine the tone and framing of the message a bit condescending. Some of the language used in the paper is not good for creating new discussions and for community building.

Reading the paper from beginning, the number of times that the responsible AI literature is mentioned is way too many. The notions of rigor that paper considers are well known and well established concepts in science, some of them for centuries. I do not really understand why this paper has to draw those concepts from the literature of responsible AI. One would guess authors are from that community and it's easier for them to use the literature of their own community. But, I think the paper can keep its open and broad view, the view described in the first paragraph, and use all the scientific literature that is available to it. In other words, the responsible AI literature needs to be mentioned, but there seems to be an unnecessary discrepancy between claiming that rigor can be different than responsible AI, yet repeatedly talking about responsible AI literature.

**Questions:**

I really do not like how the paper frames its argument in its conclusion. It seems that the paper is saying AI community has ignored the discussions under responsible AI -- now, we reframe those discussions under a new banner hoping that AI community will now consider them.

I know many people from the responsible AI community who consider themselves active members of the broader AI community. Why does not this paper consider itself part of the AI community? Why does the paper draw a line between AI community and responsible AI? I think if the paper changes its view and considers itself a member of the AI community, then it will adopt a slightly different language, a language that will be more likely to create productive discussions.

Discussion of alternative views also suffers from the same issue of paper viewing itself as an outsider. While there are pieces of work in the AI literature that might be flawed, there certainly are countless number of publications that are rigorous by all the 6 definitions put forward by the paper. Any community can strive to uphold higher standards, and in my view, it's good to write position papers for that without creating divisions.

**Alternative Position:**

Yes, and alternative positions are well-considered and addressed by the argument

**Author Identification:**

No.

**Context:**

2

**Discussion:**

3

**Ethics:**

["NO or VERY MINOR ethics concerns only"]

**Position:**

Yes, the paper argues for or against a position related to machine learning.

**Support:**

3

**Thoroughness:**

4

---

### Official Review · Reviewer_CRBv · 2025-08-06

**Significance:** 4
**Presentation:** 3
**Rating:** 7
**Confidence:** 3

**Summary:**

The work argues for more rigorous research in the field of AI. It defines five categories of rigor in this area:

 1. Epistemic rigor = articulating assumptions and background knowledge;
 2. Normative rigor = positioning the work in the context of one's beliefs, standards, values
 3. Conceptual rigor = defining the objects under investigation in the research
 4. Methodological rigor = defining the methods used, providing justification for their use
 5. Reporting rigor = clearly communicating research findings and supporting them with evidence
 6. Interpretative rigor = clearly communicating inferences made based on the results

The authors also provide a discussion on mechanism that would promote each type of rigor and give examples on where failing to do rigorous research under each category led to unreliable research findings.

**Strengths:**

1. The paper calls for more principled and, as the title suggests, more rigorous approach to AI research which is commendable. It is
Difficult to disagree that as a community, we would want our research to meet the highest possible standards, measure the right quantities of interest, and provide worthwhile conclusions.

2. The topic of the work is fairly abstract, but the authors provide very informative examples that help to pinpoint the true meaning behind the classes of rigor in AI research they distinguish.

3. Sections on how to promote each type of rigor are very helpful and provide the work with a hands-on context on how to do better research.

4. It is worthwhile that the work distinguishes so many categories of rigor since the common belief appears to be that methodological rigor precedes other types (or is the only type, as discussed by the authors). It is valuable o mention other kinds of considerations researchers should make while doing research to improve future research itself.

**Weaknesses:**

1. If there is any weakness I could point to, is that the work is sometimes written in a league that is difficult interpret. The provided examples and ways to promote rigor are helpful, as I mentioned above, but the language use is at times complicated. Some sentences are (too) long, e.g., "Reflecting on what to report about a study before the study is even conducted can help mitigate concerns about making such choices post-factum by hypothesizing after the results are known or selectively presenting only from positive results". Or "Reliance on undisclosed normative assumptions about problem statements that are contested or even debunked can be particularly misleading when a “field employs the language of procedural adherence to project a sense of certainty, objectivity, and stability”.

Minor:
- I would suggest to make Figure 1 larger, it is hard to read as of now
- There are a lot of missing citation throughout the work (or empty parentheses)

**Questions:**

1. What do you think is the most problematic aspect of the current research in terms of rigor that we should prioritize? If there is one, mentioning it could strengthen the work and provide incentive for researchers to improve their work.

**Alternative Position:**

No

**Author Identification:**

No.

**Context:**

3

**Discussion:**

3

**Ethics:**

["NO or VERY MINOR ethics concerns only"]

**Position:**

Yes, the paper argues for or against a position related to machine learning.

**Support:**

4

**Thoroughness:**

3

---

### Official Review · Reviewer_jpBf · 2025-08-08

**Significance:** 4
**Presentation:** 4
**Rating:** 9
**Confidence:** 4

**Summary:**

The paper argues that, to avoid or mitigate the undesirable consequences of lack of rigor, AI needs a broader conception of rigor that goes beyond the methodological kind typically considered.
It is claimed and substantiated that these undesirable consequences stem in many cases from aspects that precede and/or follow methodological considerations.
The authors describe six facets of rigor (only one of which is methodological), and they illustrate the points at each step using examples of real-world research.

**Strengths:**

Very well written, referenced, and argued. The examples given throughout complement the arguments effectively.
The position is important and potentially impactful. The points apply equally well to a broad range of subfields in AI.

**Weaknesses:**

No major weaknesses to report (see minor comments and suggestions in the Questions section).

**Questions:**

* Table 1 is very small. Can you make it such that the font size is more similar to the main text?
* Figure 1 is very small. Can you make it such that the font size is more similar to the main text?
* Consider using different colors and line styles for each of the larger boxes and making some room so that their lines don't come too close. This should clarify where exactly normative considerations and background knowledge overlap.
* The last part of the sentence on lines 66-69 is a bit hard to parse for this reviewer.
* Consider avoiding repetition of the blue text in the preambles of the sections on facets of rigor (i.e., mention once and then use the appropriate pronouns).
* The section on pre-registration seems to take onboard a distinction between confirmatory and exploratory research that only holds up under a particular lens which is arguably not shared across communities. See Feest, U., & Devezer, B. (2025). Toward a More Accurate Notion of Exploratory Research (And Why it Matters).
* Reading your discussion of facets of rigor, I did not see any obvious place for rigor in assessing theoretical plausibility (e.g., Adolfi et al, 2025, ICLR; Adolfi et al., 2024; Comp Brain Behav). Can these facets accommodate it?

**Alternative Position:**

Yes, and alternative positions are well-considered and addressed by the argument

**Author Identification:**

No.

**Context:**

4

**Discussion:**

3

**Ethics:**

["NO or VERY MINOR ethics concerns only"]

**Position:**

Yes, the paper argues for or against a position related to machine learning.

**Support:**

4

**Thoroughness:**

4

---

### Note · Authors · 2025-09-05

**1-11 Submit Again:**

Probably yes

**1-1 Submission Process:**

4

**1-4 Interest:**

["Panel discussions with other position paper authors", "Structured debates on controversial topics", "Mentorship programs for early-career researchers"]

**1-5 Thoughtful:**

8

**1-6 Supportive:**

8

**1-7 Technical Aspects Versus Position:**

9

**1-8 Gate Keeping:**

8

**1-9 Camera Ready Changes:**

First, we are thankful and encouraged by the thoughtful and very positive reviews. While we will do a careful editing pass and make the changes we mention below, these changes are minor and straightforward to apply.

To summarize, for the camera ready version of our paper, we will address the minor suggestions our reviewers made about the formatting of the figure and the table (e.g., regarding font size and coloring), we will also split and/or shorten longer sentences to make the text easier to parse, and we will change the convention we used to mark deleted text in quotes (i.e., the empty square brackets “[]” within quotes do not represent missing citations but are intended to mark deleted text; to avoid confusion we will instead use ellipsis in square brackets “[...]”).

We will also integrate the references suggested by the R1, which we find interesting and relevant. Specifically, we will integrate brief notes on theoretical plausibility which is relevant for both epistemic and interpretative considerations/rigor, and we will slightly expand the discussion on confirmatory versus exploratory research to recognize the underlying assumptions this distinction makes.

In response to R3, we will clarify that we do not claim that *only* responsible AI research covers aspects related to the six facets of rigor we foreground, but that we ground our discussion about these facets on work from the responsible AI community because it discusses these aspects in the context of AI research and practice (which is what our paper is concerned with). We will also edit the 2nd and 3rd sentences in the beginning of the conclusions section to further clarify that we do indeed agree and argue that responsible AI should not be cast as out of scope for AI research and practice.

**3-1 Review Response1:**

jpBf

**3-2 Reaction To Review1:**

We are really encouraged and grateful that the reviewer found our position important and impactful, and the paper well-written and well-argued. We also found the pointers that the reviewer provided to additional references interesting and really thoughtful!

We will address all the minor suggestions the reviewer made and integrate these references into the camera-ready version of the paper. While we were not aware of these references (which seem fairly recent), we find aspects related to both 1) theoretical plausibility and 2) common assumptions underlying the distinction between confirmatory and exploratory research relevant to parts of our discussion about mechanisms that can help promote epistemic and interpretative rigor.

**3-3 Review Response2:**

CRBv

**3-4 Reaction To Review2:**

We appreciate the reviewer’s thoughtful and positive review, and we are happy that they are supportive of our position and that they found the paper to be written in a way that makes it makes it valuable, informative and very helpful. To answer the reviewer’s question about which facet we should prioritize, this will generally depend on the research scope and goals. This being said, a lack of epistemic or conceptual rigor is likely to impact all other facets of rigor. For the camera-ready version of this paper, we will split and/or shorten longer sentences to make the text easier to follow and address the other two very minor comments the reviewer made. The empty square brackets “[]” within quotes do not represent missing citations but are intended to mark deleted text; to avoid confusion we will instead use ellipsis in square brackets “[...]”

Given the content of the review, we are slightly confused about the reviewer's answer to the question about the alternative position. We discuss alternative views on page 9.

**3-5 Review Response3:**

2RoQ

**3-6 Reaction To Review3:**

We thank the reviewer for closely engaging with our paper. While the reviewer may not share all of our views or fully agree with our position, we are encouraged that they overall remain positive about the paper, finding it well-written, timely and relevant.

We do not intend to be condescending. What we aimed to do was to *take a strong, clear position that encourages debates* about both how to broaden our community's conception of rigor, as well as some of the aspects the reviewer points to (e.g., that responsible AI work/community is and should be seen as part of AI work/community). Our view is that the responsible AI community is and should be seen as part of the AI community, but this does not seem to be a dominant view in the broader AI community. Regarding our backgrounds, we provide in the footnote on page 2 a brief positionality statement that provides a high-level overview of our disciplinary backgrounds. We also agree with the reviewer that the six facets of rigor we foreground are foundational to good science and we do not claim that *only* responsible AI research covers aspects related to these facets. The reason we chose to primarily draw on responsible AI research is because it specifically examines these aspects in the context of AI research and practice (which is what our position paper is concerned with). For the camera-ready version of our paper, we will make edits to clarify these points.

---

### Meta-Review · Area_Chair_C7Mw · 2025-08-31

**Rating:** 8
**Confidence:** 3

**Strengths:**

* The reviewers agree that the paper is well-written, clearly argued, and supported by strong references and effective examples that clarify its abstract topic.
* The reviewers agree that this paper will create discussion.
* The categories of rigor are interesting, and it is positive that methodological rigor is not the only category.
* The paper provides actionable advice.
* The paper is relevant and can have impact.

**Weaknesses:**

* The paper would benefit from an attempt to shorten long sentences. If different communities should understand the content, easier is better.
* The over emphasis of responsible AI literature seems out of place, or should be acknowledged at the beginning of the paper.

**Questions:**

* Would we need to weight different categories of rigor differently?
* What are the risks of such structure? What lessons did the social sciences learn from applying pre-registration as a format?

**Ethics:**

No concerns raised.

**Thoroughness:**

2

---

### Decision · Program_Chairs · 2025-09-26

Accept